# The Cryo-EM structure of pannexin 1 reveals unique motifs for ion selection and inhibition

Kevin Michalski[1†], Johanna L Syrjanen[2†], Erik Henze[1], Julia Kumpf[1], Hiro Furukawa[2]*, Toshimitsu Kawate[1]*

[1]Department of Molecular Medicine, Cornell University, Ithaca, United States; [2]WM Keck Structural Biology Laboratory, Cold Spring Harbor Laboratory, Cold Spring Harbor, United States

**Abstract** Pannexins are large-pore forming channels responsible for ATP release under a variety of physiological and pathological conditions. Although predicted to share similar membrane topology with other large-pore forming proteins such as connexins, innexins, and LRRC8, pannexins have minimal sequence similarity to these protein families. Here, we present the cryo-EM structure of a frog pannexin 1 (Panx1) channel at 3.0 Å. We find that Panx1 protomers harbor four transmembrane helices similar in arrangement to other large-pore forming proteins but assemble as a heptameric channel with a unique constriction formed by Trp74 in the first extracellular loop. Mutating Trp74 or the nearby Arg75 disrupt ion selectivity, whereas altering residues in the hydrophobic groove formed by the two extracellular loops abrogates channel inhibition by carbenoxolone. Our structural and functional study establishes the extracellular loops as important structural motifs for ion selectivity and channel inhibition in Panx1.

**\*For correspondence:**
furukawa@cshl.edu (HF);
toshi.kawate@cornell.edu (TK)

[†]These authors contributed equally to this work

**Competing interests:** The authors declare that no competing interests exist.

## Introduction

Large-pore forming channels play important roles in cell-to-cell communication by responding to diverse stimuli and releasing signaling molecules like ATP and amino acids (*Giaume et al., 2013*; *Ma et al., 2016*; *Okada et al., 2018*; *Osei-Owusu et al., 2018*). Pannexins are a family of ubiquitously expressed large-pore forming channels which regulate nucleotide release during apoptosis (*Chekeni et al., 2010*), blood pressure (*Billaud et al., 2011*; *Billaud et al., 2015*), and neuropathic pain (*Bravo et al., 2014*; *Weaver et al., 2017*; *Mousseau et al., 2018*). While pannexins have limited sequence identity with innexins (~15% identity), they have virtually no sequence similarity to other large-pore forming channels (*Panchin et al., 2000*). Among the pannexin family, pannexin 1 (Panx1) has garnered the most attention for its role as a large-pore forming channel responsible for ATP release from a variety of cell types (*Bao et al., 2004*; *Dahl, 2015*). Different kinds of stimuli have been reported to activate Panx1 including voltage, membrane stretch, increased intracellular calcium levels, and positive membrane potentials (*Bruzzone et al., 2003*; *Bao et al., 2004*; *Locovei et al., 2006*; *Wang et al., 2014*; *Chiu et al., 2018*). Panx1 is also targeted by signaling effectors, such as proteases and kinases, to permanently or temporarily stimulate channel activity (*Pelegrin and Surprenant, 2006*; *Thompson et al., 2008*; *Sandilos et al., 2012*; *Billaud et al., 2015*; *Lohman et al., 2015*). The above evidence suggests that Panx1 has a capacity to integrate distinct stimuli into channel activation leading to ATP release. Despite playing critical roles in a variety of biological processes, a mechanistic understanding of pannexin function has been largely limited due to the lack of a high-resolution structure. Here, we show the cryo-EM structure of Panx1, which reveals the pattern of heptameric assembly, pore lining residues, important residues for ion selection, and a putative carbenoxolone binding site.

## Results

### Structure determination and functional characterization

To identify a pannexin channel suitable for structure determination, we screened 34 pannexin orthologues using Fluorescence Size Exclusion Chromatography (FSEC)(*Kawate and Gouaux, 2006*). Frog Panx1 (frPanx1; 66% identical to human, *Figure 1—figure supplement 1*) displayed high expression levels and remained monodisperse when solubilized in detergent, suggesting high biochemical integrity. We further stabilized frPanx1 by truncating the C-terminus by 71 amino acids and by removing 24 amino acids from the intracellular loop between transmembrane helices 2 and 3 (*Figure 1—figure supplement 1*). This construct, dubbed 'frPanx1-ΔLC', displayed high stability in detergents and could be purified to homogeneity (*Figure 1—figure supplement 2a and b*). We verified that frPanx1 forms a functional pannexin channel by whole-cell patch clamp electrophysiology

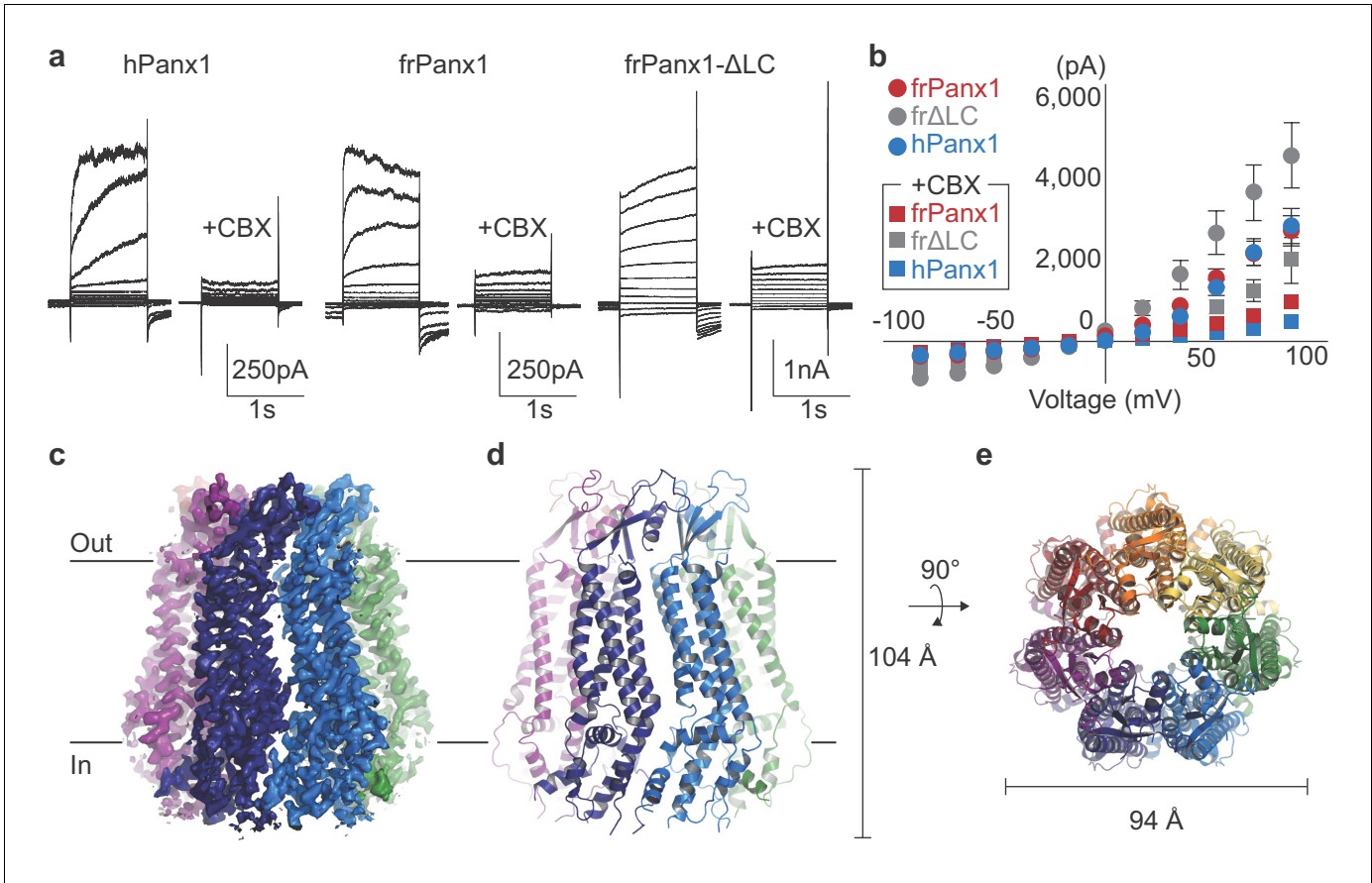

**Figure 1.** frPanx1 forms a heptameric ion channel. (a) Whole-cell patch clamp recordings from HEK 293 cells expressing hPanx1, frPanx1, and frPanx1-ΔLC. Cells were clamped at −60 mV and stepped from −100 mV to +100 mV for 1 s in 20 mV increments. To facilitate electrophysiological studies, we inserted a Gly-Ser motif immediately after the start Met to enhance Panx1 channel opening as we previously described (*Michalski et al., 2018*). CBX (100 μM) was applied through a rapid solution exchanger. (b) Current-voltage plot of the same channels shown in (a). Recordings performed in normal external buffer are shown as circles, and those performed during CBX (100 μM) application are shown as squares. Each point represents the mean of at least three different recordings, and error bars represent the SEM. (c) EM map of frPanx1-ΔLC shown from within the plane of the membrane. Each protomer is colored differently, with the extracellular side designated as 'out' and the intracellular side as 'in.' (d) Overall structure of frPanx1-ΔLC viewed from within the lipid bilayer. (e) Structure of frPanx1 viewed from the extracellular face.

The online version of this article includes the following figure supplement(s) for figure 1:

**Figure supplement 1.** Sequence alignment and structural features.

**Figure supplement 2.** Characterization of frPanx1-ΔLC.

**Figure supplement 3.** Cryo-EM image processing workflow for single particle analysis of frPanx1-ΔLC.

**Figure supplement 4.** Representative cryo-EM density of frPanx1-ΔLC.

(*Figure 1a and b*; *Figure 1—figure supplement 2e and f*). Purified frPanx1-ΔLC was reconstituted into nanodiscs composed of MSP2N2 (an engineered derivative of apolipoprotein) and soybean polar lipids, and subjected to cryo-electron microscopy (cryo-EM) and single-particle analysis (*Figure 1—figure supplement 2c and d*). We used a total of 90,185 selected particles for 3D reconstruction at 3.0 Å resolution (*Figure 1—figure supplement 3*). The map quality was sufficient for de novo model building for the majority of frPanx1-ΔLC with the exception of disordered segments of the N-terminus (residues 1–10), ECL1 (88–100), and ICL1 (157–194) (*Figure 1c*; *Figure 1—figure supplement 4*, *Video 1*, and *Table 1*).

## Overall structure and protomer features

The frPanx1-ΔLC structure revealed a heptameric assembly, which is unique among the known eukaryotic channels (*Figure 1d and e*). Other large-pore forming channels include hexameric connexins (*Maeda et al., 2009*) and LRRC8s (*Deneka et al., 2018*; *Kasuya et al., 2018*; *Kefauver et al., 2018*), and the octameric innexins (*Oshima et al., 2016*) and calcium homeostasis modulator1 (CALHM1) (*Syrjanen et al., 2020*; *Figure 2—figure supplement 1*). Our result differs from previous studies that suggest hexameric assembly of pannexin based on single channel recordings on concatemeric channels and negative stain electron microscopy (*Boassa et al., 2007*; *Wang et al., 2014*; *Chiu et al., 2017*). The heptameric assembly observed in the current study is unlikely to be caused by the carboxy-terminal truncation or intracellular loop deletion because cryo-EM images of the full-length frPanx1 also display clear seven-fold symmetry in the 2D class averages (*Figure 2—figure supplement 2a*). Furthermore, 2D class averages of hPanx1 display a heptameric assembly, but not other oligomeric states (*Figure 2—figure supplement 2b*). Thus, overall, our data suggests that the major oligomeric state of Panx1 is a heptamer. This unique heptameric assembly is established by inter-subunit interactions at three locations: 1) ECL1s and the loop between β2 and β3; 2) TM1-TM1 and TM2-TM4 interfaces; and 3) α9 helix and the surrounding α3 and α4 helices, and the N-terminal loop from the neighboring subunit (*Figure 2—figure supplement 3*). Notably, the majority of residues mediating these interactions are highly conserved (e.g. Phe67 and Tyr111; *Figure 1—figure supplement 1*).

The overall protomer structure of Panx1 resembles that of other large-pore forming channels including connexin, innexins, and LRRC8. Like other large-pore forming channels, each Panx1 protomer harbors four transmembrane helices (TM1-4), two extracellular loops (ECL1 and 2), two intracellular loops (ICL1 and 2), and an amino (N)-terminal loop (*Figure 2a and b*). The transmembrane helices of Panx1 are assembled as a bundle in which the overall helix lengths, angles, and positions strongly resemble the transmembrane arrangements observed in other large-pore channels (*Figure 2c*). In contrast, Panx1 has no similarity in transmembrane arrangement to another group of large-pore channels, CALHMs whose protomers also contain four transmembrane helices (*Choi et al., 2019*; *Syrjanen et al., 2020*; *Figure 2—figure supplement 1*). Structural features in the Panx1 ECL1 and ECL2 domains are conserved among large-pore channels despite limited sequence similarity (*Figure 2d–g*; *Figure 2—figure supplement 1*). For example, the Panx1 ECL1 and ECL2 are joined together by two conserved disulfide bonds (Cys66 with Cys267, Cys84 with Cys248) in addition to several β-strands. ECL1 also contains an alpha-helix that extends towards the central pore and forms an extracellular constriction of the permeation pathway. While much of the transmembrane domains and extracellular loops show similarities to other large-pore forming channels, the Panx1 intracellular domains are structurally unique (*Figure 2—figure supplement 1*). ICL1 and ICL2, for example, together form a bundle of helices that make contact with the N-terminus. The N-terminal loop of Panx1 forms a constriction of the permeation pathway and extends towards the intracellular region. The first ~10 amino acids of the N-terminus are

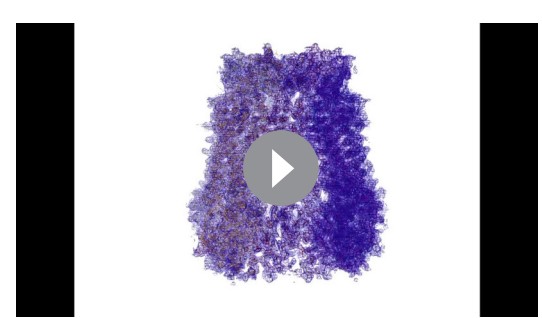

**Video 1.** Cryo-EM density of frPanx1-ΔLC. The model is shown as wire representation and fit into the corresponding density contoured at σ = 3.0. Each domain is colored differently and Tryp74 and Arg75 are labeled in the close-up view.
https://elifesciences.org/articles/54670#video1

**Table 1.** Cryo-EM data collection, refinement and validation statistics.

| | frPanx- ΔLC (EMD-21150) (PDB: 6VD7) |
|---|---|
| Data collection and processing | |
| Magnification | 130,000 |
| Voltage (kV) | 300 |
| Electron exposure (e–/Å$^2$) | 57.2 |
| Defocus range (μm) | 1.2–2.8 |
| Pixel size (Å) | 1.07 |
| Symmetry imposed | C7 |
| Initial particle images (no.) | 297374 |
| Final particle images (no.) | 90185 |
| Map resolution (Å)<br>FSC threshold | 3.02<br>0.143 |
| Refinement | |
| Initial model used (PDB code) | de novo |
| Model resolution (Å)<br>FSC threshold | 3.29<br>0.5 |
| Model resolution range (Å) | 3–6 |
| Map sharpening $B$ factor (Å$^2$) | −90 |
| Model composition<br>Non-hydrogen atoms<br>Protein residues<br>Ligands | 16506<br>2079<br>0 |
| CC map vs. model (%) | 0.85 |
| R.m.s. deviations<br>Bond lengths (Å)<br>Bond angles (°) | 0.008<br>0.759 |
| Validation<br>MolProbity score<br>Clashscore<br>Poor rotamers (%) | 1.92<br>5.96<br>0.78 |
| Ramachandran plot<br>Favored (%)<br>Allowed (%)<br>Disallowed (%) | 88.32<br>11.68<br>0 |

disordered in our structure, but these residues might play a role in ion permeation or ion selectivity (*Wang and Dahl, 2010*).

## Ion permeation pathway and selectivity

The Panx1 permeation pathway spans a length of 104 Å, with constrictions formed by the N-terminal loop, Ile58, and Trp74 (*Figure 3a and b*). The narrowest constriction is surrounded by Trp74 located on ECL1 (*Figure 3c*). Trp74 is highly conserved among species including hPanx1 (*Figure 1—figure supplement 1*). Because Panx1 has been previously characterized as an anion selective channel (*Ma et al., 2012*; *Romanov et al., 2012*; *Chiu et al., 2014*), we wondered if positively charged amino acids around the narrowest constriction formed by Trp74 may contribute to anion selectivity of the channel. Interestingly, Arg75 is situated nearest to the tightest constriction of the permeation pathway (*Figure 3d*). We hypothesized that Arg75 might be a major determinant of anion selectivity of Panx1 channels in the open state. To assess whether Arg75 contributes to anion selectivity, we generated a series of point mutations at this position on hPanx1 and compared their reversal potentials (Erev) in asymmetric solutions using whole-cell patch clamp electrophysiology (*Figure 3e* and *Figure 3—figure supplement 1*). We kept sodium chloride (NaCl) constant in the pipette solution

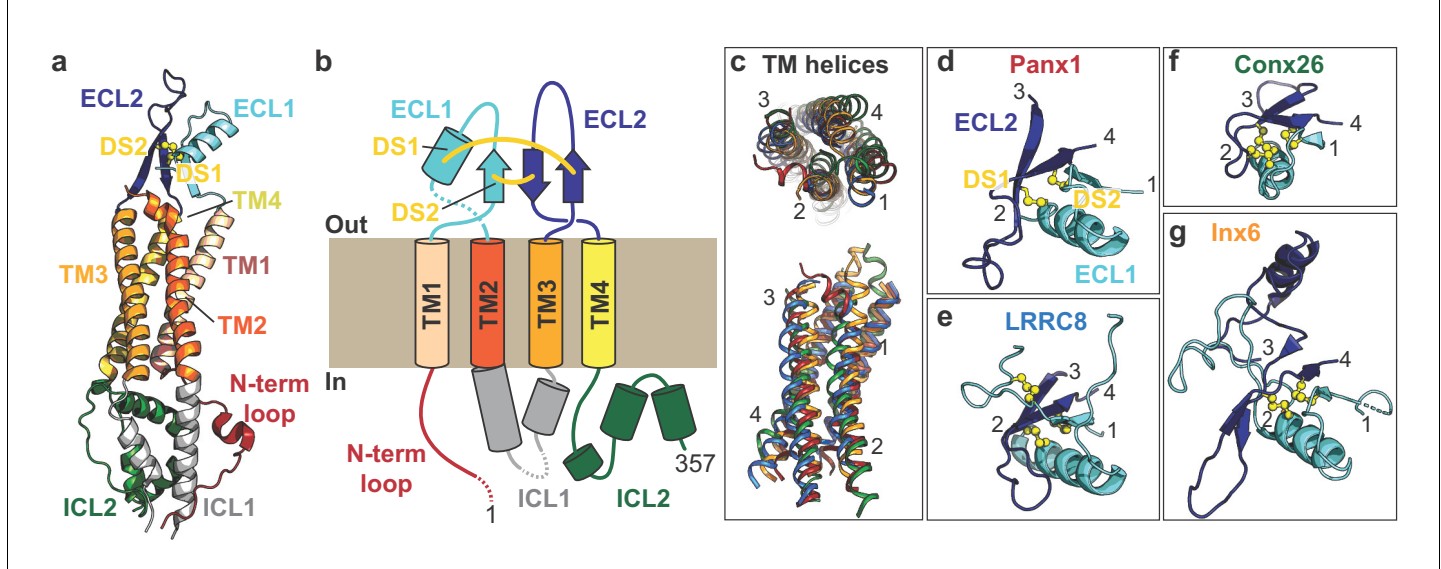

**Figure 2.** Subunit architecture of frPanx1. (a) Structure of the frPanx1 protomer. Each domain is colored according to the cartoon scheme presented in (b). (c) Superimposition of the transmembrane helices from frPanx1 (red), connexin-26 (green), innexin-6 (orange), and LRRC8 (blue) shown top-down from the extracellular side (top) or from within the plane of the membrane (bottom). (d-g) Cartoon representation of the extracellular loops of large pore forming channels. ECL1 is colored in light blue, and ECL2 is colored in dark blue, and disulfide bridges are shown as yellow spheres. These domains are viewed from the same angle (from top) as shown in the top panel in (c).

The online version of this article includes the following figure supplement(s) for figure 2:

**Figure supplement 1.** Comparison of frPanx1 with other large pore channels.
**Figure supplement 2.** 2D classes of full-length frog and human pannexin 1.
**Figure supplement 3.** Inter-subunit interactions.

while varying the extracellular solution. When treated with the large anion, gluconate (Gluc⁻), $E_{rev}$ shifted to +26 mV, suggesting the channel is more permeable to Cl⁻ than to Gluc⁻. When exposed to the large cation, N-methyl-D-glucamine (NMDG⁺), $E_{rev}$ remained close to 0 mV, suggesting that Na⁺ and NMDG⁺ equally (or do not) permeate Panx1. These results are consistent with Panx1 being an anion-selective channel. The Arg75Lys mutant maintains the positive charge of this position, and displayed $E_{rev}$ values comparable to WT. Removing the positive charge at this position, as shown by the Arg75Ala mutant, diminished Cl⁻ selectivity as the $E_{rev}$ in NaGluc remained near 0 mV. Interestingly, the $E_{rev}$ in NMDGCl shifted to −22 mV, suggesting the channel had lost anion selectivity and Na⁺ became more permeable than NMDG⁺. A charge reversal mutant, Arg75Glu, shifted the $E_{rev}$ in NaGluc to −16 mV and in NMDGCl to −45 mV, indicating that Gluc⁻ became more permeable to Cl⁻. Overall, these results support the idea that the positively charged Arg75 plays a role in anion selectivity of Panx1.

We next wondered if introducing a charge at position 74 might alter ion selectivity of Panx1 channels. Interestingly, both Trp74Arg and Trp74Glu mutants become less selective to anions and more permeable to Na⁺ (*Figure 3e*). These results suggest that introducing a charge at this position disrupts the natural ion selectivity of Panx1 channels but that position 74 itself does not control ion selectivity. We observed that the distance between the guanidino group of Arg75 and the benzene ring of Trp74 from an adjacent subunit is ~4 Å, suggesting that these two residues likely participate in an inter-subunit cation-π interaction key to Panx1 ion selectivity (*Figure 3f*). To test this hypothesis, we generated Trp74Ala and Trp74Phe mutations and measured $E_{rev}$ potentials. Trp74Ala showed a marked decrease in Cl⁻ permeability and an increase in Na⁺ permeability, despite preservation of the positive charge at Arg75. A more conservative mutation, Trp74Phe, still disrupted ion selectivity, suggesting that proper positioning of the benzene ring at position 74 is important for anion selection. Altogether, our data suggests that anion selectivity is only achieved when Trp74 and Arg75 form a cation-π interaction. Given that our structure has disordered and truncated regions in the N-terminus, ICL1, and ICL2, it is possible that additional ion selectivity or gating regions exist in

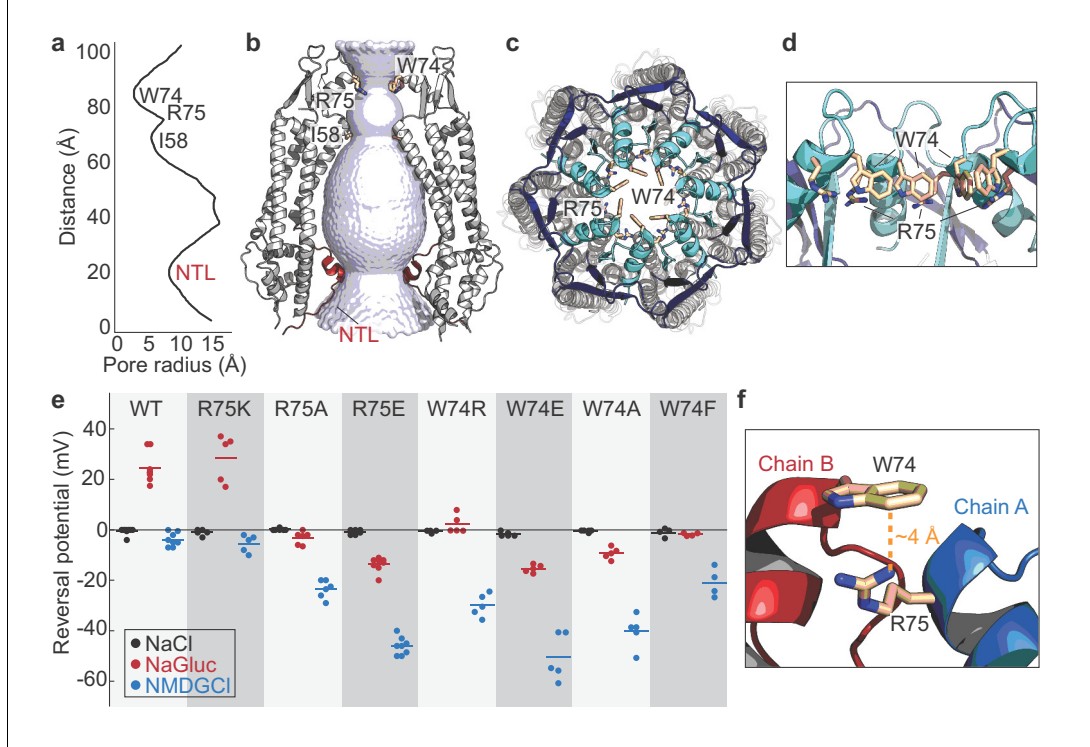

**Figure 3.** Permeation and ion selectivity of Panx1 channels. (**a**) HOLE (*Smart et al., 1996*) diagram demonstrating constrictions along the permeation pathway. NTL; N-terminal loop. (**b**) Surface representation of the internal space along the molecular 7-fold axis running through the center of frPanx1. The surface was generate using HOLE. (**c and d**) Top view facing the extracellular side (**c**) or side view (**d**) of frPanx1, with ECL1 shown in light blue and ECL2 in dark blue. Trp74 and Arg75 are shown as sticks. (**e**) Reversal potentials of various hPanx1 ion selectivity mutants. Each point represents the Erev measured in NaCl (black), NaGluc (red), or NMDGCl (blue), and bars represent the mean values. I-V curves were obtained by a ramp protocol from −80 mV to +80 mV. (**f**) Close-up view of the Trp74-Arg75 interaction at the interface of protomer A (blue) and B (red).

The online version of this article includes the following figure supplement(s) for figure 3:

**Figure supplement 1.** Representative traces of the ramp recordings.

**Figure supplement 2.** Electrostatic surface potential of the ion permeation pathway.

the full-length channel. For example, the N-termini of LRRC8 and connexins perform an important role in ion selectivity (*Kyle et al., 2008*; *Kronengold et al., 2012*; *Kefauver et al., 2018*). It is possible that the N-terminus of Panx1 is mobile and may further constrict the permeation pathway. Another possibility is that the electrostatic potential along the pore pathway contributes to the ion selectivity. Interestingly, both cytoplasmic and extracellular entrances of the permeation pathway are mostly basic, suggesting that non-permeant cations may be excluded from the pore (*Figure 3—figure supplement 2*). In contrast, the region underneath the W74 constriction is highly acidic, supporting the idea that anions may be selected around this area.

## CBX action mechanism

We have previously demonstrated that CBX, a potent nonselective inhibitor of Panx1, likely acts through a mechanism involving ECL1 (*Michalski and Kawate, 2016*). In these experiments, mutations at a number of residues in ECL1 rendered Panx1 less sensitive to CBX-mediated channel inhibition. Mapping such residues in the Panx1 structure revealed that they are clustered proximal to the extracellular constriction by Trp74, in a groove formed between ECL1 and ECL2 (*Figure 4a and b*). This supports our previous speculation that CBX is an allosteric inhibitor, not a channel blocker (*Michalski and Kawate, 2016*).

Given that this hydrophobic groove is formed also by residues in ECL2, we wondered if residues in ECL2 might also play a role in CBX-mediated inhibition. We mutated selected residues in ECL2 of hPanx1 to cysteines and measured channel activity before and after CBX application. We found that

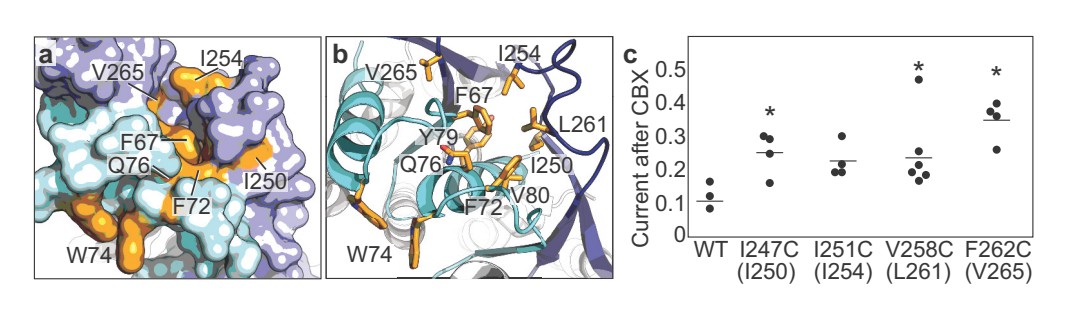

**Figure 4.** CBX action requires residues from both ECL1 and ECL2. (**a and b**) Surface (**a**) and cartoon (**b**) representations of the frPanx1 ECL1 (light blue) and ECL2 (dark blue), with potential CBX-interacting residues shown in orange. (**c**) Quantification of whole-cell currents from hPanx1 mutants when treated with CBX (100 µM). Mutants are numbered according to the hPanx1 sequence while the mutants in parenthesis are the corresponding residues in frPanx1. Recordings were performed by stepping to +100 mV in the absence or presence of CBX, and each point represents the normalized current amplitude during the CBX application. Bars represent the mean value from each mutant. Asterisks indicate significance of $p<0.05$ determined by one-way ANOVA followed by Dunnett's test comparing WT to each mutant (F262C: $p=0.0007$; I247C: $p=0.0471$; V258C: $p=0.0363$).

mutations at Ile247, Val258, and Phe262 (hPanx1 numbering) diminished CBX-sensitivity (*Figure 4c*). These data suggest that both ECL1 and ECL2 play important roles in inhibition of Panx1 by CBX. Although we do not have a cryo-EM structure complexed to CBX at this point, we speculate that CBX inhibits Panx channels by binding between ECL1 and ECL2 and 'locking' the conformation of gate forming ECL1 in favor of channel closure.

## Discussion

The frPanx1-ΔLC structure uncovered a unique heptameric assembly of a large-pore channel that harbors an extracellular constriction formed by Trp74 and Arg75. These residues are located on ECL1 and face toward the central pore of the channel and thus, are situated to regulate channel function. Mutagenesis studies at these positions revealed that both residues play pivotal roles in ion selection. Unlike the LRRC8A anion channel, however, the positively charged Arg75 does not seem to form a canonical selectivity filter. Instead, the guanidino group of Arg75 likely mediates a cation-π interaction with Trp74 in the neighboring subunit, which seems to control ion selection. One possible ion selection mechanism is that this cation-π interaction stabilizes the inter-subunit interactions, which in turn creates an electrostatic environment that favors anion permeation. Another possibility is that tight inter-subunit interactions in the extracellular domain are necessary to form an ion selectivity filter in the missing region in our current model (e.g. N-terminus or C-terminal domain).

Which functional state does our model represent? Based on the lack of channel activity at 0 mV (*Figure 1—figure supplement 2e and f*), our current structure may represent a closed conformation. This is supported by the existence of a highly acidic region near Trp74 (*Figure 3—figure supplement 2*), which may serve as a barrier for anions to permeate. However, given that the narrowest constriction at Trp74 is ~10 Å wide, it is possible that the structure actually represents an open conformation. Indeed, the +GS version of frPanx1-ΔLC shows larger leak currents (*Figure 1a and b*), suggesting that the C-terminal truncation may promote channel opening while lack of the N-terminal modification renders it closed. If the conformation of the N-terminus in frPanx1-ΔLC is somehow compromised during purification or reconstitution into nanodiscs, it is possible that our structure may actually look closer to the +GS version. While further studies are necessary to define the functional state of our current structure, the weak EM density in the N-terminal region leaves the possibility that frPanx1-ΔLC may be representing an open state.

We found that ECL1 and ECL2 interact to each other and form a potential CBX binding pocket. Both ECL1 and ECL2 may undergo movement based on conformational alterations of the TMDs and cytoplasmic domains. For example, it is conceivable that movement of the TMDs caused by membrane stretch or voltage, or changes in the cytoplasmic domain triggered by caspase cleavage may be coupled to conformational rearrangements in the extracellular domain. The major role of the

extracellular domain in pannexin function is strongly supported by our experimental results demonstrating that mutating Trp74 and Arg75, as well as surrounding residues in ECL1 and ECL2, alter channel properties including ion selectivity. Furthermore, we previously demonstrated that application of CBX to mutants at Trp74 (e.g. to Ala, Ile, Lys) potentiates voltage-dependent channel activity (*Michalski and Kawate, 2016*), which indicates that CBX likely acts as an allosteric inhibitor rather than a channel blocker.

In contrast to the extracellular domain, roles of the intracellular domain remain elusive. While the C-terminal domain has been demonstrated to play important roles in Panx1 channel gating (*Sandilos et al., 2012*), our study neither confirms or refutes this mechanism as half of this domain is missing in our current structure. Likewise, the first 10 residues in the N-terminus are disordered, making it challenging to understand how these residues tune the activity of Panx1 channel (*Michalski et al., 2018*). Given their important roles in channel gating, it is possible that the unmodeled N-terminal region may interact with the deleted region of the C-terminal domain. It is also possible that these domains may form a channel gate. In contrast to these domains, the deleted residues in ICL-1 (between Gly171 and Lys194) seems to play a minimal role in channel gating. We surveyed 23 different deletion constructs (in which each variant harbored a different deletion length and position) and among these, all deletions constructs showed voltage-dependent channel activity via whole-cell patch clamp, with the exception of a construct in which the entire region between Lys155 and Lys194 was removed. We also tested these deletion constructs using FSEC and found that all functional constructs were properly assembled into heptamers. The above evidence indicates that the deleted region in ILC-1 plays an insignificant role in channel gating. The EM density in this region was weak and could not be modeled, indicating a high degree of conformational flexibility.

In conclusion, our frPanx1-ΔLC structure provides an important atomic blueprint for dissecting functional mechanisms of Panx1. While we did not observe a gate-like structure in the current cryo-EM map, the missing domains, especially the N-terminal loop and the C-terminal domain, may serve as a channel gate on the intracellular side of the channel. Further structure-based experiments such as cysteine accessibility and molecular dynamics simulations will facilitate our understanding of how this unique large-pore channel functions.

## Materials and methods

### Key resources table

| Reagent type (species) or resource | Designation | Source or reference | Identifiers | Additional information |
|---|---|---|---|---|
| Gene (*Xenopus tropicalis*) | frPanx1 | Synthesized by Genscript | NCBI Reference Sequence: NP_001123728.1 | Frog pannexin-1 gene sequence |
| Gene (*Homo sapiens*) | hPanx1 | Synthesized by Genscript | NCBI Reference Sequence: NP_056183.2 | Human pannexin-1 gene sequence |
| Cell line (*Homo sapiens*) | HEK293T cells | ATCC | Cat#: CRL-3216, RRID: CVCL_0045 | |
| Cell line (*Spodoptera frugiperda*) | Sf9 cells | ATCC | Cat#: CRL-1711, RRID: CVCL_0549 | |
| Recombinant DNA reagent | pIE2 hPanx1 | DOI: 10.1085/jgp.201711804 | | Mammalian expression vector for electrophysiology presented in *Figure 1—figure supplement 1 and 2* |
| Recombinant DNA reagent | pIE2 hPanx1 +GS | DOI: 10.1085/jgp.201711804 | | Mammalian expression vector for electrophysiology presented in *Figures 1*, *3* and *4* |
| Recombinant DNA reagent | pIE2 frPanx1 | This paper | | Mammalian expression vector for electrophysiology presented in *Figure 1—figure supplement 1 and 2* |

*Continued on next page*

*Continued*

| Reagent type (species) or resource | Designation | Source or reference | Identifiers | Additional information |
|---|---|---|---|---|
| Recombinant DNA reagent | pIE2 frPanx1 +GS | This paper | | Mammalian expression vector for electrophysiology presented in *Figure 1* |
| Recombinant DNA reagent | pIE2 frPanx1-ΔLC | This paper | | Mammalian expression vector for electrophysiology presented in *Figure 1—figure supplement 1 and 2* |
| Recombinant DNA reagent | pIE2 frPanx1-ΔLC +GS | This paper | | Mammalian expression vector for electrophysiology presented in *Figure 1* |
| Recombinant DNA reagent | pC-NG-FB7 frPanx1-ΔLC | This paper | | Insect cell/baculovirus expression construct |
| Recombinant DNA reagent | pC-NG-FB7 frPanx1 | This paper | | Insect cell/baculovirus expression construct |
| Recombinant DNA reagent | pC-NG-FB7 hPanx1 | This paper | | Insect cell/baculovirus expression construct |
| Peptide, recombinant protein | MSP2N2 | doi: 10.1016/S0076-6879 (09)64011–8 | | nanodisc expression construct |
| Commercial assay or kit | Fugene 6 | Promega | Cat#: E2691 | |
| Chemical compound, drug | Carbenoxolone | Sigma | Cat#: C4790 | |
| Chemical compound, drug | C12E8 | Anatrace | Cat#: APO128 | |
| Chemical compound, drug | DDM | Anatrace | Cat#: D310 | |
| Chemical compound, drug | Soybean polar lipid extract | Avanti | Cat#: 541602 | |
| Software, algorithm | cisTEM | DOI: 10.7554/eLife.35383 | RRID: SCR_016502 | |
| Software, algorithm | Warp | DOI: 10.1038/s41592-019-0580-y | | |
| Software, algorithm | Coot | DOI: 10.1107/S0907444904019158 | RRID: SCR_014222 | |
| Software, algorithm | PHENIX | DOI: 10.1107/S09074449052925 | RRID: SCR_014224 | |
| Software, algorithm | Axon pClamp 10.5 | Axon (Molecular Devices) | RRID: SCR_011323 | |

## Cell line generation

HEK293 (CRL-1573) cell lines were purchased from the American Type Culture Collection (ATCC, Manassas, VA), and therefore were not further authenticated. The mycoplasma contamination test was confirmed to be negative at ATCC.

## Purification of frPanx1-ΔLC

frPanx1 (NP_001123728.1) was synthesized (Genscript) and cloned into the BamHI/XhoI sites of pCNG-FB7 vector containing a C-terminal Strep-tag II (WSHPQFEK). Amino acids from the IL1 and IL2 were removed by standard PCR strategies, and the BamHI site was also removed by quickchange mutagenesis. The full length frPanx1 and hPanx1 (NP_056183.2; synthesized by Genscript) were also subcloned into pCNG-FB7 vectors by standard PCR. Sf9 cells were infected with high titer baculovirus (20–25 mL P2 virus/L cells) at a cell density of 2.5–3.0 $\times$ 10$^6$ cells/ mL and cultured at 27℃ for 48 hr. Cells were collected by centrifugation, washed once with PBS, and lysed by nitrogen cavitation

(4635 cell disruption vessel; Parr Instruments) at 600 psi in PBS containing leupeptin (0.5 µg/mL), aprotinin (2 µg/mL), pepstatin A (0.5 µg/mL), and phenylmethylsulfonyl fluoride (0.5 mM). Broken cells were centrifuged at 12,000 x g for 10 min, and membranes were collected by ultracentrifugation at 185,000 x g for 40 min. Membranes were suspended and solubilized in PBS containing 1% $C_{12}E_8$ (Anatrace) for 40 min, followed by ultracentrifugation at 185,000 x g for 40 min. Solubilized material was incubated with StrepTactin Sepharose High-Performance resin (GE Healthcare) for 40 min in batch. Resin was collected onto a gravity column (Bio-Rad), washed with 10 column volumes of wash buffer (150 mM NaCl, 100 mM Tris-HCl pH 8.0, 1 mM EDTA, 0.5 mM $C_{12}E_8$), and eluted with five column volumes of wash buffer supplemented with 2.5 mM desthiobiotin. Eluted protein was concentrated and further purified on a Superose 6 10/300 Increase column (GE Healthcare) with 150 mM NaCl, 10 mM Tris pH 8.0, 0.5 mM DDM as the running buffer. Peak fractions were collected and pooled. All steps were performed at 4°C or on ice.

## Reconstitution into nanodiscs

MSP2N2 apolipoprotein was expressed and purified as described previously (*Ritchie et al., 2009*), and the N-terminal His tag was cleaved off using TEV protease prior to use. To incorporate frPanx1 into nanodiscs, soybean polar extract, MSP2N2 and frPanx were mixed at final concentrations of 0.75, 0.3 and 0.3 mg/ml, respectively. The mixture was incubated end-over-end for 1 hr at 4°C, followed by detergent removal by SM2 Bio-Beads (Bio-Rad). The supernatant and wash fractions were collected after an overnight incubation (~12 hr) and further purified by size exclusion chromatography using a Superose 6 10/300 column in 20 mM Tris-HCl pH 8.0, 150 mM NaCl, 1 mM EDTA. Peak fractions were pooled and concentrated to 3 mg/mL.

## Cryo-EM sample preparation and image collection

frPanx1 in nanodiscs or hPanx1 in n-Dodecyl-β-D-Maltopyranoside (DDM; Anatrace) were applied to glow-discharged lacey carbon-coated copper grids (Electron Microscopy Services). The grids were blotted for 4 s with blot force 7 at 85% humidity at 15°C, and plunge frozen into liquid ethane using a Vitrobot Mark IV (Thermo Fisher). All data were collected on a FEI Titan Krios (Thermo Fisher) operated at an acceleration voltage of 300 keV. For frPanx1-ΔLC, a total of 2034 images were collected at 130 k magnification with a pixel size of 1.07 Å in electron counting mode. Each micrograph was composed of 32 frames collected over 4 s at a dose of 1.79 e / Å$^2$/frame and a total exposure per micrograph of 57.3 e / Å$^2$. Data were collected using EPU software (FEI). For full-length frPanx1 in nanodiscs, a total of 574 images were collected at 130 k magnification with a pixel size of 1.06 Å in electron counting mode. Each micrograph was composed of 50 frames collected over 10 s at a dose of 1.4 e / Å$^2$/frame. The total exposure per micrograph was 70 e / Å$^2$. Data were collected using SerialEM (*Schorb et al., 2019*). Data for full-length hPanx1 in DDM were collected in a similar fashion.

## Cryo-EM image processing and single particle analysis

Warp was used for aligning movies, estimating the CTF and particle picking for frPanx1-ΔLC and full-length hPanx1. For full-length frPanx1, movie alignment and CTF estimation were performed using the program Unblur and CTFFind, respectively, within the cisTEM package (*Grant et al., 2018*). 2D classification, ab-initio 3D map generation, 3D refinement, 3D classification, per particle CTF refinement and B-factor sharpening were performed using the program cisTEM (*Grant et al., 2018*). The single particle analysis workflow for frPanx1-ΔLC is shown in *Figure 1—figure supplement 3*. De novo modeling was performed manually in Coot (*Emsley and Cowtan, 2004*). The final model was refined against the cryo-EM map using PHENIX real space refinement with secondary structure and Ramachandran restraints (*Adams et al., 2010*). The FSCs were calculated by phenix. mtriage. Data collection and refinement statistics are summarized in Extended data *Table 1*.

## Electrophysiology

HEK293 cells were plated onto 12 mm glass coverslips (VWR) in wells of a six-well plate and transfected 24 hr later with 500–800 ng plasmid DNA using FUGENE 6 (Promega) according to the manufacturer's instructions. Recordings were performed ~16–24 hr later using borosilicate glass micropipettes (Harvard Apparatus) pulled and polished to a final resistance of 2–5 MΩ. Pipettes

were backfilled with (in mM) 147 NaCl, 10 EGTA, 10 HEPES pH 7.0 with NaOH. Patches were obtained in external buffer composed of (in mM) 147 NaCl, 10 HEPES pH 7.3 with NaOH, 13 glucose, 2 KCl, 2 CaCl$_2$, 1 MgCl$_2$. A rapid solution exchange system (RSC-200; Bio-Logic) was used to perfuse cells with CBX or various salt solutions. Currents were recorded using an Axopatch 200B amplifier (Axon Instruments), filtered at 2 kHz (Frequency Devices), digitized with a Digidata 1440A (Axon Instruments) with a sampling frequency of 10 kHz, and analyzed with the pClamp 10.5 software (Axon Instruments). For voltage step recordings, Panx1-expressing cells were held at −60 mV and stepped to various voltage potentials for 1 s in 20 mV increments before returning to −60 mV. For ramp recordings, cells were held at −60 mV, and ramped between −100 mV and + 100 mV over 3 s duration.

## Acknowledgements

We thank the members of the Kawate and the Furukawa lab for discussions. We also thank D Thomas and M Wang for managing the cryo-EM facility and the computing facility at Cold Spring Harbor Laboratory, respectively. This work was supported by the National Institutes of Health (GM114379 to TK; NS113632 to HF; GM008267 to KM, EK, and JK; GM008267 to KM), Robertson funds at Cold Spring Harbor Laboratory, Doug Fox Alzheimer's fund, Austin's purpose, and Heartfelt Wing Alzheimer's fund (to HF). JLS is supported by the Charles H Revson Senior Fellowship in Biomedical Science.

## Additional information

### Funding

| Funder | Grant reference number | Author |
|---|---|---|
| National Institutes of Health | GM114379 | Toshimitsu Kawate |
| National Institutes of Health | NS113632 | Hiro Furukawa |
| National Institutes of Health | GM008267 | Kevin Michalski<br>Erik Henze<br>Julia Kumpf |
| Cold Spring Harbor Laboratory | Robertson funds | Hiro Furukawa |
| Doug Fox Alzheimer's fund | | Hiro Furukawa |
| Austin's Purpose | | Hiro Furukawa |
| Heartfelt WingAlzheimer's Fund | | Hiro Furukawa |
| Charles H. Revson Foundation | Senior Fellowship in Biomedical Science | Johanna L Syrjanen |

The funders had no role in study design, data collection and interpretation, or the decision to submit the work for publication.

### Author contributions

Kevin Michalski, Data curation, Formal analysis, Validation, Investigation, Methodology, Writing - review and editing; Johanna L Syrjanen, Data curation, Software, Formal analysis, Investigation, Visualization, Methodology, Writing - review and editing; Erik Henze, Data curation, Formal analysis, Investigation, Writing - review and editing; Julia Kumpf, Data curation, Formal analysis, Investigation, Methodology; Hiro Furukawa, Resources, Data curation, Software, Formal analysis, Supervision, Validation, Investigation, Methodology, Writing - review and editing; Toshimitsu Kawate, Conceptualization, Resources, Data curation, Formal analysis, Supervision, Funding acquisition, Validation, Investigation, Visualization, Methodology, Writing - original draft, Project administration, Writing - review and editing

## Author ORCIDs

Julia Kumpf  http://orcid.org/0000-0002-3813-1255
Toshimitsu Kawate  https://orcid.org/0000-0002-5005-2031

## Decision letter and Author response

Decision letter https://doi.org/10.7554/eLife.54670.sa1
Author response https://doi.org/10.7554/eLife.54670.sa2

# Additional files

## Supplementary files

• Transparent reporting form

## Data availability

Cryo EM data and the pannexin model has been deposited in PDB under the accession code 6VD7.

The following dataset was generated:

| Author(s) | Year | Dataset title | Dataset URL | Database and Identifier |
|---|---|---|---|---|
| Syrjanen JL, Michalski K, Furukawa H, Kawate T | 2020 | Cryo-EM structure of *Xenopus tropicalis* pannexin 1 channel | https://www.rcsb.org/structure/6VD7 | RCSB Protein Data Bank, 6VD7 |

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
