## [Decision Letter]

**Acceptance summary:**

This is an important manuscript describing the first near atomic resolution structure of a pannexin channel, a family of large pore channels thought to be important for the movement of ions and small molecules such as ATP across cellular membranes. The mechanisms by which the activity of pannexin channels are regulated are poorly understood, yet these channels are thought to play important roles in innate immunity, neuronal signaling, apoptosis, cancer and ischemia. Having a structure of the channel will enable further mechanistic studies on how this protein plays important biological roles. By combining data from cryo-EM and electrophysiology, this manuscript describes the first structure of a pannexin channel at near-atomic resolution and it sheds light on the determinants of anion selectivity, gating and pharmacology. The data appears to be of high quality and the major conclusions of the manuscript are sound. The reviewers agreed that the work is a strong candidate for publication in *eLife*, and the authors have done an excellent job of revising the manuscript to address the concerns of the reviewers.

**Decision letter after peer review:**

Thank you for submitting your article "The Cryo-EM Structure of a Pannexin 1 Channel Reveals an Extracellular Gating Mechanism" for consideration by *eLife*. Your article has been reviewed by Kenton Swartz as the Senior Editor, a Reviewing Editor, and three reviewers. The following individuals involved in review of your submission have agreed to reveal their identity: Raimund Dutzler (Reviewer #2); Ming Zhou (Reviewer #3).

The reviewers have discussed the reviews with one another and the Reviewing Editor has drafted this decision to help you prepare a revised submission.

Summary:

This is an important manuscript describing the first near atomic resolution structure of a pannexin channel, a family of large pore channels thought to be important for the movement of ions and small molecules (e.g. ATP) across cellular membranes. The reviewers thought this was an excellent manuscript that is appropriate for publication in *eLife*. As is spelled out in the reviews below, all three reviewers have suggestions for improving the manuscript in revision. Most of the issues they raise can be readily addressed by careful revision of the text to openly discuss unresolved questions or to tone down some conclusions (e.g. the potential CBX binding site). We ask that you prepare a revised version that addresses the issues raised to the best of your ability.

*Reviewer #1:*

This manuscript presents for the first time the structure of a pannexin channel. This is a timely and very important finding for the field that will serve as a basis for understanding and studying pannexin-1 channel function.

I think it is important to note that in the current (and frenetic) race of structural biologists to determine channel structures, the main concern for physiologists is whether these structures really represent physiological conformations of the channel. I believe the authors prove, at least partially, that this is the case with the frog pannexin-1 channel. The structural interpretations that the extracellular loop 1, in particular, residues at positions W74 and R75 regulate ionic selectivity are nicely supported by their functional data. The functional data supporting inhibition by carbenoxolone at extracellular region are complementary, but less demonstrative.

The major concern with this work is with respect to some forced interpretations that do not properly address the limitations of the pannexin-1 structure and require proper clarification. This refers mainly to the lack of reliable structural data related for the cytosolic domains N-terminus (NT), intracellular loops (ICL) and C-terminus (CT), which have been shown to be crucial in gating and permeation of these channels. This needs to be properly acknowledged in the manuscript.

Another manuscript describing the pannexin-1 structures from frog and human was published a few days after the current manuscript in BioRxiv. They share similar structural results, and both lack structural information for the N-terminal and C-terminal domain.

Essential revisions:

1) As stated by the authors, the NT region has been shown to regulate gating and permeation in pannexin-1 channels. In this presented pannexin-1 structure the first 10 residues of the NT region are not resolved. The remaining residues are lining the pore as shown for other large pore channels including connexin, innexin and LRRC8 channels. Importantly, the intracellular loops in the aforementioned structures are critical for stabilizing the N-terminal region. This has been more recently demonstrated for connexin hemichannels. To solve the pannexin-1 structure, the authors had to remove 24 amino acids from the first ICL. How the deletion of this segment could alter the interactions between ICL1, ICL2, and NT, and the potential implications in channel gating or selectivity need to be discussed.

2) The structure does not contain a significant part of the C-terminal region, which has been shown to play an important role in the gating of human, rat and mouse Pannexin 1 channels. The lack of the CT in pannexin-1 leads to constitutively open channels in these species, even at negative potential. How is therefore explained that the frog pannexin-1 structure lacking the CT region corresponds to a closed channel conformation as the authors propose?

3) The authors suggest that the motif formed by the residues W74 and R75 is an extracellular gate in pannexin-1 channels. They referred to this motif in the text as a unique mechanism of gating, which gives the impression of a unique gate. If W74 is the only gate, one would predict that mutations in W74 (such as W74A) will disrupt the gate and produce a leaky channel, due to the substitution of an amino acid with a short side chain. This does not seem to be observed by the authors. If the mutations do not lead to a leak current, it is likely that there is a second gate for atomic ions.

4) Pannexin channel currents for different constructs shown in Figure 1B and Figure 1—figure supplement 2F are not normalized by channel levels. This implies that currents of frPanx1, hPanx1, and frPanx1-ΔLC are not comparable.

5) Figure 1—figure supplement 2F suggests that the channels used to determine the cryo-EM structure (i.e., those formed by frPanx1-ΔLC) are not functional. As noted, the authors needed to add a GS motif in the NT region to increase voltage-sensitivity. This reinforces the idea of a role for the NT in gating.

6) The Discussion section is incomplete. I think that many of the concerns in this review need to be addressed by the authors in this section.

7) The authors conclude that "our structural and functional study establishes the extracellular loops as the unique structural determinants for channel gating and inhibition in Panx 1…". Considering the above issues this sentence is incorrect. It must be also noted that the inhibition mechanism proposed by the authors is only ascribable to CBX. I think that modeling/docking of CBX with the pannexin-1 structure will reinforce the experimental data and provide a more solid conclusion about the mechanism of blockade by CBX.

*Reviewer #2:*

By combining data from cryo-EM and electrophysiology, the work describes the first structure of a Pannexin channel at high resolution and it sheds light on the determinants of anion selectivity. This is an excellent manuscript that provides important insight into an important ion channel. The data appears to be of high quality and the major conclusions of the manuscript are sound. I thus think that the work is a strong candidate for publication in *eLife*.

Essential revisions:

– The cryo-EM data is of high quality and it supports the structural claims made in the manuscript. The ring of aromatic residues at the extracellular constriction and the close-by arginine residues are particularly interesting, and the proposed cation-π electron interaction is supported by the data (considering the limited resolution of the map). Whereas the role of both residues for shaping the selectivity properties of the protein have been nicely demonstrated in patch clamp recordings, the evidence for the same region to act as gate is to some degree vague and speculative since it has not been demonstrated that the same residues would provide that largest energy-barrier that obstructs permeation in the closed conformation.

– Given the proposal that different region contribute to anion selectivity it would be interesting to know more about the electrostatics within the pore region.

– I wonder whether there is evidence that the described Arg residues contribute to the observed voltage dependence of currents. Although the authors have shown traces of recordings of the mutants to demonstrate shifts in the reversal potential, possible changes in the voltage dependence of gating could be better documented. I expect in their structure, the positively charged Arg would be located outside the electric field.

– I wonder whether the authors have additional evidence to exclude a potential role of CBX as pore blocker. Is the effect of CBX reversible?

– Are small currents after addition of CBX reflect incomplete occupancy of the blocker or residual conduction of the CBX-bound channel and are currents in mutants a consequence of decreased potency or increased conductance of the blocked channel (given the large amount of data presented in the study such experiments might be focus of a future study).

– In the electrophysiology section, it would be important to show some comparative data from mock-transfected cells.

– Was the change in the junction potential in asymmetric solutions in the selectivity experiments calculated to be significant and was it corrected?

*Reviewer #3:*

The manuscript from the Kawate lab titled "The cryo-EM structure of a pannexin 1 channel reveals an extracellular gating mechanism" reports the structure of a pannexin 1 from frog at about 3.0 Å resolution and functional studies that tested structure-based hypotheses on ion selectivity and inhibitor binding site. The work is significant in that this is the first pannexin structure, and that the structure revealed a heptameric assembly different from other large-pore forming channels such as connexins, innexins and LRRC8. The initial FSEC and excellent biochemical work led to identification of a stable construct suitable for structural studies. The procedures for cryo-EM seem straightforward, and the quality of the density map seems consistent with the claimed resolution.

I feel that the experiments are all solid and interpretations are sound. I have two comments.

First, the study on CBX is a very good follow up on a previous JGP paper by the same lab, and it provides further evidence that extracellular loop 1 harbors the inhibitor. However, short of a CBX-pannexin complex, one could argue that the inhibitor may bind elsewhere and induce a propagated effect that is mediated by the extracellular loop.

Second, I see utility of using ion permeation as a read out for testing certain aspects of pannexin, but I do not quite understand the motivation of studying ion selectivity in pannexin. If the main function of a pannexin is to form a large pore to allow release of ATP, would it be more relevant to study permeation of ATP and its gating on a proteoliposome assay? Is there any indication that anion permeation is relevant in physiology?

---

## [Author Response]

Summary:This is an important manuscript describing the first near atomic resolution structure of a pannexin channel, a family of large pore channels thought to be important for the movement of ions and small molecules (e.g. ATP) across cellular membranes. The mechanisms by which the activity of pannexin channels are regulated are unknown, yet these channels are thought to play important roles in innate immunity, neuronal signaling, apoptosis, cancer and ischemia. Having a structure of the channel will enable further mechanistic studies on how this protein plays important biological roles. The reviewers thought this was an excellent manuscript that is appropriate for publication in eLife. As is spelled out in the reviews below, all three reviewers have suggestions for improving the manuscript in revision. Most of the issues they raise can be readily addressed by careful revision of the text to openly discuss unresolved questions or to tone down some conclusions (e.g. the potential CBX binding site). We ask that you prepare a revised version that addresses the issues raised to the best of your ability.

We thank the reviewers for the valuable comments. As suggested, we have toned down uncertain conclusions and added more discussion throughout the manuscript, especially on the limitations of our current study. We believe our revised and retitled manuscript addresses the concerns raised by the reviewers. Below we summarize our point-by-point answers/comments to the suggestions made by the reviewers.

Reviewer #1:[…]Essential revisions:1) As stated by the authors, the NT region has been shown to regulate gating and permeation in pannexin-1 channels. In this presented pannexin-1 structure the first 10 residues of the NT region are not resolved. The remaining residues are lining the pore as shown for other large pore channels including connexin, innexin and LRRC8 channels. Importantly, the intracellular loops in the aforementioned structures are critical for stabilizing the N-terminal region. This has been more recently demonstrated for connexin hemichannels. To solve the pannexin-1 structure, the authors had to remove 24 amino acids from the first ICL. How the deletion of this segment could alter the interactions between ICL1, ICL2, and NT, and the potential implications in channel gating or selectivity need to be discussed.

We created 23 different deletions in ICL1 between K155 and K194. Except for the construct that lacks this entire region, all other constructs showed voltage-dependent channel activity in our whole-cell patch clamp experiments. We also investigated these deletion constructs using FSEC and found that the functional constructs were properly assembled into heptamers. Based on these results, we think the deleted region in the loop has insignificant contribution in channel gating. Indeed, the EM density in this region was weak and we could not model the residues between K155 and K194. The NT region on the other hand does seem to play a crucial role in channel gating, as we previously demonstrated (Michalski et al., 2018). It is possible that the unmodeled NT region may interact with the deleted region of the CTD. We included this information in the Discussion section.

2) The structure does not contain a significant part of the C-terminal region, which has been shown to play an important role in the gating of human, rat and mouse Pannexin 1 channels. The lack of the CT in pannexin-1 leads to constitutively open channels in these species, even at negative potential. How is therefore explained that the frog pannexin-1 structure lacking the CT region corresponds to a closed channel conformation as the authors propose?

In our hands, CT truncated Panx1 (1-355 in human) does not open constitutively. This is independent of species or intracellular loop deletion. Because frPanx1-ΔLC is closed at 0 mV (Figure 1—figure supplement 2), we initially thought that the current structure represents a closed state. However, the +GS version of frPanx1-ΔLC shows larger leak currents (Figure 1), leaving a possibility that the CT truncation does promote channel opening but lack of the NT modification somehow renders it closed. If the conformation of the NT in frPanx1-ΔLC is compromised during purification or reconstitution into nanodiscs, it is possible that our structure may actually look closer to the +GS version, which may then represent an open state. We included this discussion in our revised manuscript.

3) The authors suggest that the motif formed by the residues W74 and R75 is an extracellular gate in pannexin-1 channels. They referred to this motif in the text as a unique mechanism of gating, which gives the impression of a unique gate. If W74 is the only gate, one would predict that mutations in W74 (such as W74A) will disrupt the gate and produce a leaky channel, due to the substitution of an amino acid with a short side chain. This does not seem to be observed by the authors. If the mutations do not lead to a leak current, it is likely that there is a second gate for atomic ions.

We agree with the reviewer. Because we do not have sufficient data to confirm W74's role as a channel gate, we removed the phrase "extracellular gate" from our manuscript.

4) Pannexin channel currents for different constructs shown in Figure 1B and Figure 1—figure supplement 2F are not normalized by channel levels. This implies that currents of frPanx1, hPanx1, and frPanx1-ΔLC are not comparable.

That is true. We do not intend to compare those constructs quantitatively, but aim to qualitatively demonstrate that all three constructs present voltage-dependent channel activity when the N-termini are modified with the addition of Gly-Ser.

5) Figure 1—figure supplement 2F suggests that the channels used to determine the cryo-EM structure (i.e., those formed by frPanx1-ΔLC) are not functional. As noted, the authors needed to add a GS motif in the NT region to increase voltage-sensitivity. This reinforces the idea of a role for the NT in gating.

We agree that the NT region plays essential roles in pannexin channel gating. Unfortunately, our current EM structure falls short of proposing a possible mechanism, as the first 10 amino acids are not modelled. As suggested by the reviewers, we expanded the Discussion section to clarify the limitation of our studies.

6) The Discussion section is incomplete. I think that many of the concerns in this review need to be addressed by the authors in this section.

Thank you for the comments. Our revised Discussion section should address the concerns raised by the reviewers.

7) The authors conclude that "our structural and functional study establishes the extracellular loops as the unique structural determinants for channel gating and inhibition in Panx 1…". Considering the above issues this sentence is incorrect. It must be also noted that the inhibition mechanism proposed by the authors is only ascribable to CBX. I think that modeling/docking of CBX with the pannexin-1 structure will reinforce the experimental data and provide a more solid conclusion about the mechanism of blockade by CBX.

We used "unique" in a sense that it is unusual for an ion channel to have tryptophan residues surrounding the most constricted region of the pore. However, we agree with the reviewer that the sentence is misleading. We therefore toned down its potential role as a gating machinery throughout the manuscript. We also agree that further studies are necessary to clarify the action mechanism of CBX.

Reviewer #2:[…]Essential revisions:– The cryo-EM data is of high quality and it supports the structural claims made in the manuscript. The ring of aromatic residues at the extracellular constriction and the close-by arginine residues are particularly interesting, and the proposed cation-π electron interaction is supported by the data (considering the limited resolution of the map). Whereas the role of both residues for shaping the selectivity properties of the protein have been nicely demonstrated in patch clamp recordings, the evidence for the same region to act as gate is to some degree vague and speculative since it has not been demonstrated that the same residues would provide that largest energy-barrier that obstructs permeation in the closed conformation.

Thank you for the comments, which we agree. As described above, we have toned down the potential extracellular gating mechanism.

– Given the proposal that different region contribute to anion selectivity it would be interesting to know more about the electrostatics within the pore region.

Thank you for the suggestion. The electrostatic potential map revealed that the ion permeation pathway is mostly basic except for a highly acidic region near the constriction formed by W74. This charge distribution is consistent with an anion channel with a selectivity machinery located near W74. We now include an extra figure (Figure 3—figure supplement 2) and discussion in the text (subsection “Ion permeation pathway and selectivity”).

– I wonder whether there is evidence that the described Arg residues contribute to the observed voltage dependence of currents. Although the authors have shown traces of recordings of the mutants to demonstrate shifts in the reversal potential, possible changes in the voltage dependence of gating could be better documented. I expect in their structure, the positively charged Arg would be located outside the electric field.

We did observe various degrees of voltage dependence from different R75 mutants. However, it is challenging to assess the contribution of the positively-charged side chain in voltage gating, as the interaction with W74 also seems to affect voltage dependence. One clear thing is that the positive charge at this position is not necessary for voltage-dependent channel activity, as R75A or even R75E give rise to voltage-dependent currents.

– I wonder whether the authors have additional evidence to exclude a potential role of CBX as pore blocker. Is the effect of CBX reversible?

Yes, the effect of CBX is reversible. We previously demonstrated that voltage-dependent channel activity of W74A, W74I, or W74K is potentiated by CBX (Michalski et al., 2016). In combination with our systematic mutagenesis studies, CBX is more likely to be an allosteric inhibitor than a channel blocker.

– Are small currents after addition of CBX reflect incomplete occupancy of the blocker or residual conduction of the CBX-bound channel and are currents in mutants a consequence of decreased potency or increased conductance of the blocked channel (given the large amount of data presented in the study such experiments might be focus of a future study).

This is an interesting point. Our previous single channel recordings suggest that CBX reduces the channel open probability without changing the conductance (Michalski et al., 2018). This suggests that the residual currents after CBX application reflect incomplete occupancy. But we agree that further experiments are necessary to clarify this.

– In the electrophysiology section, it would be important to show some comparative data from mock-transfected cells.

We have already published several responses from mock-transfected cells that confirm the channel activities in our lab indeed come from Panx1 (Michalski et al., 2016 and Michalski et al., 2018).

– Was the change in the junction potential in asymmetric solutions in the selectivity experiments calculated to be significant and was it corrected?

We corrected junction potentials experimentally using an agar bridge.

Reviewer #3:[…]I feel that the experiments are all solid and interpretations are sound. I have two comments.First, the study on CBX is a very good follow up on a previous JGP paper by the same lab, and it provides further evidence that extracellular loop 1 harbors the inhibitor. However, short of a CBX-pannexin complex, one could argue that the inhibitor may bind elsewhere and induce a propagated effect that is mediated by the extracellular loop.

We appreciate the comments. We agree with the reviewer that CBX may bind outside of the tested residues. And yes, CBX-pannexin complex structure will be very helpful.

Second, I see utility of using ion permeation as a read out for testing certain aspects of pannexin, but I do not quite understand the motivation of studying ion selectivity in pannexin. If the main function of a pannexin is to form a large pore to allow release of ATP, would it be more relevant to study permeation of ATP and its gating on a proteoliposome assay? Is there any indication that anion permeation is relevant in physiology?

While ATP release has been widely studied, anion permeation through pannexin channels has also been implicated in physiology (e.g. Thompson et al., 2008). The suggested in vitro ATP-release assay would be interesting.